# Facile Preparation of Organo-Modified ZnO/Attapulgite Nanocomposites Loaded with Monoammonium Glycyrrhizinate via Mechanical Milling and Their Synergistic Antibacterial Effect

**Fangfang Yang [1], Yameng Song [1,2], Aiping Hui [1], Yuru Kang [1], Yanmin Zhou [3] and Aiqin Wang [1,\*]**

[1] Key Laboratory of Clay Mineral Applied Research of Gansu Province, Center of Eco-Material and Green Chemistry, Lanzhou Institute of Chemical Physics, Chinese Academy of Sciences, Lanzhou 730030, China; yangff@licp.cas.cn (F.Y.); ymsongym@163.com (Y.S.); aphui1215@163.com (A.H.); yurukang@licp.cas.cn (Y.K.)

[2] College of Chemistry and Chemical Engineering, Northwest Normal University, Lanzhou 730070, China

[3] College of Animal Science and Technology, Nanjing Agricultural University, Nanjing 210095, China; zhouym6308@163.com

\* Correspondence: aqwang@licp.cas.cn; Tel.: +86-931-496-8118; Fax: +86-931-496-8019

**Abstract:** In this study, monoammonium glycyrrhizinate (MAG) was introduced into cetyltrimethyl ammonium bromide (CTAB)-modified ZnO/attapulgite (APT) via a mechanical process to form performance-enhanced antibacterial nanocomposites (MAG/C–ZnO/APT). The APT supported ZnO nanocomposite (ZnO/APT) was prepared by a conventional precipitation method, and 20–50 nm of globular ZnO nanoparticles were uniformly decorated on APT nanorods. The FTIR and zeta potential analyses demonstrated that modification by CTAB facilitated the loading of MAG into ZnO/APT by H-bonding and electrostatic interactions. Antibacterial evaluation results indicate that MAG/C–ZnO/APT nanocomposites with CTAB and MAG doses of 2.5% and 0.25%, respectively, exhibited synergistically enhanced inhibitory activities against *Escherichia coli*, *Staphylococcus aureus*, *Enterococcus faecalis*, *Pseudomonas aeruginosa*, methicillin-resistant *Staphylococcus aureus* and extended-spectrum β-lactamases *Escherichia coli*, with minimum inhibitory concentrations of 1, 0.1, 0.25, 5, 0.1, and 2.5 mg/mL, respectively, which are better than those of ZnO/APT, C–ZnO/APT and MAG. Moreover, the nanocomposites had low cytotoxicity on human normal cell line L-O2. Therefore, this study provided a more effective strategy to extend the antibacterial spectrum and strengthen the inhibitory effects of antibiotic-free materials to address increasingly serious situations of microbial infection.

**Keywords:** Monoammonium glycyrrhizinate; CTAB; ZnO; Attapulgite; nanocomposite; antibacterial

## 1. Introduction

For a long period of time, maleficent bacteria have caused great harm to people's lives and production, and the pandemic threat of the novel Coronavirus pandemic has intensified concerns about human health. Recently one of the most concerns is the drug resistance of bacteria, which is mainly induced by abuse of antibiotics in animal feedings for preventing infection and improving the growth of intensive farming animals [1,2]. The rapid development of drug-resistant bacteria makes people have to stop using antibiotic growth promoters that was proposed by many countries as a ban [3,4], thus posing a serious challenge to develop novel effective and safe alternative antibacterial materials to replace antibiotics used for the management of animal health.

Inorganic nanoparticles (NPs) have attracted much attention in the antibacterial field due to their excellent activities and unique antibacterial mechanism. Zinc oxide (ZnO) is one of the most widely used inorganic nanomaterials, which is demonstrated to have good effects in preventing diarrhea and promoting growth in animals [5]. However,

high dose addition of ZnO for the desired result is often required, which could cause Zn toxicity, Zn accumulation in the environment and even the development of drug resistance [6]. Therefore, currently, one question was raised as to how to apply ZnO safely or in a controlled safe dose but still achieve as good of an effect as the antibiotic growth promoter alternatives after the ban on antibiotics. A Previous study demonstrated that incorporating ZnO NPs onto attapulgite (APT) is an effective approach to strengthen the antibacterial performance of ZnO NPs due to the dispersion of ZnO NPs on APT nanorods, which can form small-sized ZnO NPs [7]. APT, a natural nanoscale clay mineral, has features of porous structure, rod-like morphology and abundant surface-active groups [8], thus drawing more attention in biomedical applications and especially being regarded as favorable building blocks for the design and construction of nanostructured composite materials by loading metal or metal oxide NPs [9], organic antibacterial agents [10] and essential oils [11]. Loading NPs on APT could obtain a more advantageous morphology for enhanced antibacterial effects by reducing particle agglomeration. However, the loading amounts of NPs have much effect on the particle size, that is, high loading will lead to large particles, even forming large microstructures [12]. Our previous studies investigated the modification method by the introduction of organic agents to enhance the antibacterial activity of ZnO/APT nanocomposite with a low loading of ZnO NPs and achieved good results [13].

Plant-derived natural active components have great potential in antibacterial applications due to their many advantages, such as low toxicity, extensive resources and lack of pollution to the environment [14–16]. Monoammonium glycyrrhizinate (MAG), a derivative of glycyrrhizic acid (GA) obtained from the roots of licorice, has remarkable bacterial inhibition and other beneficial pharmacological activities, such as anti-inflammatory activity, antiviral activity, autoxidation function and anti-gastric ulcer effects [17–19]. Additionally, GA and its derivatives have been widely used in the pharmaceutical and food industries, giving favorable information about their safety [20]. MAG has not been reported to inhibit the growth of bacteria, whereas GA has been demonstrated to have inhibitory activity against extended-spectrum β-lactamase-producing *K. pneumoniae isolates* with a minimum bactericidal concentration of 10 µg/mL [21], and have antibacterial effects against *Helicobacter pylori* from peptic ulcer patients [22]. Zhao et al. found that GA could inhibit the growth of gram-positive *Staphylococcus aureus* (*S. aureus*), but had no effect on gram-negative *Escherichia coli* (*E. coli*) [23]. Thus, such a natural ingredient attracted our attention and it is expected that using MAG to modify ZnO/APT nanocomposite will achieve enhanced antibacterial activities. However, another problem that should be considered is how to load MAG into the carrier. Cai et al. [10] have prepared a quaternary phosphonium salt modified APT nanocomposite, which exhibited controlled release of quaternary phosphonium salt for a specific-targeting antibacterial effect and lower cytotoxicity than pure quaternary phosphonium salt. Modification of clay minerals using the quaternary ammonium/phosphonium surfactants has been studied extensively [24]. The functional groups of –NH$_2$, –COO and –OH presented in MAG could interact with organic molecules by H-bonds [25]. Therefore, using the quaternary ammonium/phosphonium surfactants as a link to loading MAG onto the carrier will be a favourable strategy to be expected. In addition, the excellent antibacterial activity of such surfactants can not be ignored [26].

Herein, the aim of the study is to incorporate MAG into an organo-modified ZnO/APT by a mechanical milling process to construct the highly efficient and broad-spectrum antibacterial nanocomposites (MAG/C–ZnO/APT), and the construction process is illustrated in Scheme 1. APT served as a support to disperse ZnO NPs, and organic modification of ZnO/APT by CTAB improved the interfacial interactions of MAG with ZnO/APT and also enhanced the antibacterial capacity by strengthening the contact of the material with bacteria. The introduction of MAG enhanced the susceptibility of ZnO/APT to grampositive bacteria. Moreover, the mechanical milling process is simple and low-cost and is a technology for the large-scale production of nanocomposites. The obtained nanocomposite

exhibited favourable antibacterial performance against gram-positive and gram-negative bacteria including drug-resistant strains, as well as low cytotoxicity at low concentrations, which make it promising to be used in animal feedings as the substitution of antibiotics.

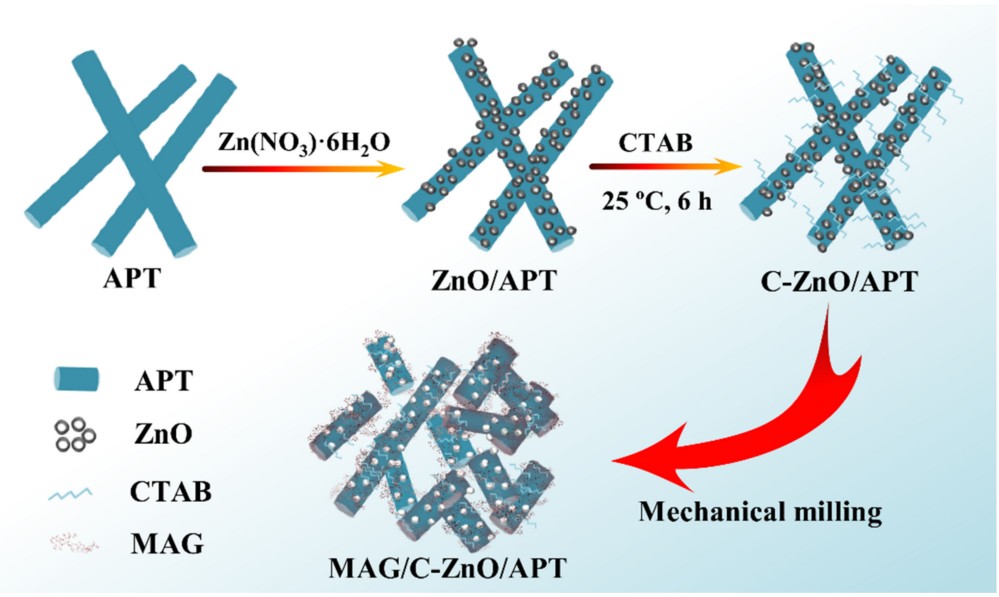

**Scheme 1.** Schematic illustration of the preparation process of MAG/C–ZnO/APT nanocomposites.

## 2. Materials and Methods

### 2.1. Materials

Attapulgite (APT) was obtained from Jiangsu Sinitic Biological Technology Co. Ltd., Jiangsu, China, and the main compositions consisted of CaO (0.65%), $Al_2O_3$ (11.79%), MgO (9.78%), $SiO_2$ (58.79%), $K_2O$ (1.39%) and $Fe_2O_3$ (6.29%), as determined by X-ray fluorescence. Monoammonium glycyrrhizinate (MAG, ≥30%) was provided by Gansu Fanzhi Pharmaceutical Co., Ltd., Gansu, China. Sodium hydroxide (NaOH, ≥96%) and sodium dodecyl sulfate (SDS, ≥99.0%) were purchased from Tianjin Kermel Chemical Reagent Co., Ltd., Tianjin, China. Zinc nitrate hexahydrate ($Zn(NO_3)_2 \cdot 6H_2O$, ≥99%) was supplied by Xilong Scientific Co., Ltd. (Guangzhou, China). Cetyltrimethyl ammonium bromide (CTAB, ≥99%) was purchased from Chengdu Cologne Chemicals Co., Ltd., Sichuan, China. Nutrient agar medium was obtained from Qingdao Hope Bio-Technology Co., Ltd., Shandong, China. 3-(4,5-Dimethylthiazol-2-yl)-2,5-diphenyltetrazolium bromide (MTT, >98%) was purchased from BioFroxx Co., LTD, Einhausen, Germany. Dimethyl sulfoxide (DMSO, ≥99.7%) was obtained from Beijing Voke Biotechnology Co., LTD, Beijing, China. Human normal cell line L-O2 and the special culture medium for L-O2 were purchased from Procell Life Science & Technology Co., Ltd., Wuhan, China. The chemicals were directly used without further purification.

### 2.2. ZnO/APT Nanocomposite Preparation and Modification

ZnO/APT nanocomposite was prepared according to the method reported by Hui et al., with minor modifications [13]. Firstly, APT (12.5 g) was fully dispersible in deionized water (100 mL), $Zn(NO_3)_2 \cdot 6H_2O$ (6.875 g) was dissolved in the above aqueous suspension and then treated by an ultrasonic for 30 min. After that, SDS (0.1875 g) was added, and then an aqueous solution of NaOH (5 wt.%) was dropped in the above mixture to adjust the pH to 7, which was stirred for 30 min at room temperature. Afterwards, the white precipitate in the mixture was separated by centrifugation at 4000× *g* for 10 min, washed three times and dried for 15 h in an oven at 80 °C. Finally, the dried product was milled and calcined at 400 °C for 3 h in an air atmosphere obtaining a ZnO/APT nanocomposite. The ZnO/APT nanocomposite was then modified by different amounts of CTAB. CTAB

with mass ratios of 0.1%, 0.25%, 0.5%, 1% and 2.5% equivalent to APT were first dissolved in 100 mL distilled water and then 5.0 g APT was slowly added. After stirring continuously for 6 h at room temperature, the reaction mixtures were filtered and washed three times with distilled water. Finally, the CTAB-modified ZnO/APT nanocomposites were dried in an air oven at 60 °C for 2 h. The resultant samples were labeled C–ZnO/APT-0.1%, C–ZnO/APT-0.5%, C–ZnO/APT-1% and C–ZnO/APT-2.5%.

### 2.3. MAG/C–ZnO/APT Nanocomposite Preparation

MAG/C–ZnO/APT nanocomposites were prepared by incorporating MAG into the CTAB-modified ZnO/APT nanocomposite under a mechanical milling process. A mass of 0.0125 g of MAG equivalent to 0.25% of APT was mixed with 5 g C–ZnO/APT samples and ground in a mortar-type milling apparatus (CRINOER, MG100, China) at 60 rpm for 30 min. The resultant samples were labeled MAG/C–ZnO/APT-0.1%, MAG/C–ZnO/APT-0.5%, MAG/C–ZnO/APT-1% and MAG/C–ZnO/APT-2.5% according to the amount of CTAB added to ZnO/APT.

### 2.4. Characterization

A field emission scanning electron microscope (FESEM, JSM-6701F, JEOL, Akishima, Japan) was used to observe the microscopic morphologies with the samples dispersed uniformly on copper and treated by gold spraying, and the element distribution was obtained from FESEM equipped with energy dispersive X-ray spectrometers (EDX). The microscopic morphology of ZnO/APT was further analyzed using a JEM-2100 high-resolution transmission electron microscope (HRTEM, JEOL, Akishima, Japan). The specific surface area based on the BET method in $N_2$ adsorption was determined at 77 K using an ASAP 2020 instrument (Micromeritics, Norcross, GA, USA). The Fourier Transform Infrared spectroscopy was undertaken on a Thermo Nicolet 6700 spectrophotometer (FTIR, Thermo Fisher Scientific, Waltham, MA, USA) in the range of 4000–400 $cm^{-1}$. Thermogravimetric analysis was carried out with a STA8000 thermal analyzer (TGA, PerkinElmer, Billerica, MA, USA), operating at 50–800 °C with a heating rate of 10 °C/min in an $N_2$ atmosphere. The zeta potentials were analyzed with a ZEN3600 zeta potentiometer (Malvern Panalytical, Malvern, UK) with the samples dispersed in diluted water at pH 6.8 and kept at a concentration of 0.5 wt.%.

### 2.5. Determination of the Minimum Inhibitory Concentration (MIC)

The minimum inhibitory concentration (MIC) was used to assay the antibacterial activities of the nanocomposites. The strains of *E. coli*, *S. aureus*, MRSA, ESBL-*E. coli*, *E. faecalis* and *P. aeruginosa* were from clinical isolates. Firstly different amounts of materials were mixed with 20 mL medium to get the LB agar plates with sample concentrations of 5, 2.5, 1, 0.5, 0.25, 0.1, and 0.05 mg/mL. After the medium cooled, 1 μL of $10^4$ CFU/mL fresh bacterial suspensions was inoculated into the plates at three different positions. Then the inoculated plates were put in an incubator and cultivated at 37 °C for 24 h. The MIC values were referring to the lowest concentration to prevent the visible growth of bacterial strains. Positive and negative control groups were set in this study. The tests were repeated at least three times.

### 2.6. Cytotoxicity Assay

MAG/C–ZnO/APT-2.5% was evaluated for in vitro cytotoxic activity on human normal cell line L-O2 using the MTT assay. Cells treated with different concentrations of material leachate with 3.75, 2.5, 1.875, 1.25, 0.625 and 0 mg/mL were seeded on 96-well plates and cultured in a humid atmosphere with 5% $CO_2$ at 37 °C for 24 h. After that MTT (5 mg/mL in PBS) was added to each well and incubated for an additional 4 h. Then, DMSO was added to the well and kept for 2 h, the optical density of various concentrations was read using a spectrophotometer at 490 nm.

The experiments were performed in triplicates. The percentage cell viability was then calculated with respect to the control (cells incubated without antibacterial agents) as follows:

$$Cell viability\% = \frac{OD_{sample}}{OD_{Control}} \times 100 \tag{1}$$

*2.7. Statistical Analysis*

The statistical analysis was conducted by SPSS programs, and all the data were repeated and expressed as the mean ± SD. The results were analyzed by one-way ANOVA followed by Student's *t*-test. Values of $p < 0.05$ were considered statistically significant.

**3. Results and Discussion**

*3.1. Formation and Characterization of MAG/C–ZnO/APT Nanocomposites*

SEM images of ZnO/APT and MAG/C–ZnO/APT-2.5% are shown in Figure 1a–i. It can be seen from Figure 1a–c that ZnO/APT displayed the rod-like morphology, and ZnO NPs with small sizes of approximately 20–50 nm are decorated on APT nanorods, which corresponds with the results from TEM (Figure S1). After being modified by CTAB, the rod-like morphology was maintained for ZnO/APT with ZnO NPs anchored on the nanorods of APT, indicating that the introduction of CTAB had no effects on the structure of ZnO/APT (Figure 1d–f). However, as seen from Figure 1g–i, which shows the microscopic morphologies of MAG/C–ZnO/APT-2.5%, some rods of ZnO/APT were broken up and the aspect ratio decreased, which was caused by the imposed mechanical force that could break the rods by shear and extrusion actions [27]. The EDX spectrum of MAG/C–ZnO/APT-2.5% nanocomposite is displayed in Figure 1j. The signals of Si, Mg, Al and Fe signify the existence of APT, and the signals of Zn and O could be detected, manifesting ZnO NPs presented in the nanocomposite. Furthermore, the elements C, N and O also indicate the successful incorporation of CTAB and MAG into the MAG/C–ZnO/APT nanocomposite. In addition, the XRD patterns (Figure 2) of APT, ZnO/APT, C–ZnO/APT-2.5% and MAG/C–ZnO/APT-2.5% provide more information about the formation and structure of the nanocomposites. In the pattern of APT, apart from the characteristic diffraction peaks of APT, peaks of the associated minerals including quartz and dolomite can be found. In the pattern of ZnO/APT, the new characteristic diffraction peaks corresponded to the characteristic hexagonal wurtzite structure of ZnO can be observed [28], demonstrating that ZnO NPs formed on the surface of APT and the average size of ZnO NPs calculated using the Scherrer method is 20 nm. The patterns of C–ZnO/APT and MAG/C–ZnO/APT-2.5% show that incorporation of CTAB and MAG had no significant effect upon the structure of ZnO/APT.

The FTIR spectra of MAG, ZnO/APT, MAG/ZnO/APT and MAG/C–ZnO/APT nanocomposites are shown in Figure 3a. In the spectra of MAG, the stretching vibrations of the –OH group was observed at 3389 cm$^{-1}$, and the C–H antisymmetric stretching modes in alkanes appeared at 2969, 2927 and 2855 cm$^{-1}$. The bands in the range of 1703–1373 cm$^{-1}$ are ascribed to the characteristic absorptions of MAG, which presented the stretching vibrations of –COO (1703 cm$^{-1}$), –C=O (1614 cm$^{-1}$), and the bending vibration of –NH$_2$ (1508 cm$^{-1}$) [22]. When MAG was loaded into ZnO/APT, the peak around 3435 cm$^{-1}$ (OH groups) became sharper than that in the spectra of ZnO/APT, indicating that MAG is loaded onto ZnO/APT possibly by the hydrogen bonds. The spectra of C–ZnO/APT are shown in Figure S2, where the stretching vibration peaks of methyl and methylene can be seen at 2954, 2925 and 2854 cm$^{-1}$, which appeared, and strengthened, with increasing amounts of CTAB, showing that the ZnO/APT surface was combined with the organic groups. This result indicated that ZnO/APT was successfully modified by CTAB. It was demonstrated that the functional groups of –NH$_2$, –COO and –OH presented in MAG could interact with organic molecules by H-bonds [25], thus MAG could be immobilized on C–ZnO/APT by forming H-bonding. Additionally, after incorporation of MAG, the absorption bands observed at 1428 cm$^{-1}$ in the patter of ZnO/APT had a more shift than

that in C–ZnO/APT, which may be caused by methyl and methylene deformation as a result of the incorporation of MAG into C–ZnO/APT. The changes in the zeta potential of MAG/C–ZnO/APT and C–ZnO/APT nanocomposites are shown in Figure 3b and Figure S3, respectively. It is observed that the zeta potential of ZnO/APT modified by CTAB exhibited a positive increase, that is, the adsorbed CTAB molecules can influence the surface charge and, consequently, the zeta potential of APT [29,30], which is significant to improvement of the antibacterial performance of ZnO/APT. After immobilization of MAG on C–ZnO/APT, the zeta potential had not changed much, suggesting that MAG molecules bond to C–ZnO/APT by the electrostatic interactions. The positive increase in the zeta potential values of ZnO/APT resulted in the enhancement of adsorption capacity to MAG molecules and is beneficial for the immobilization of MAG in C–ZnO/APT.

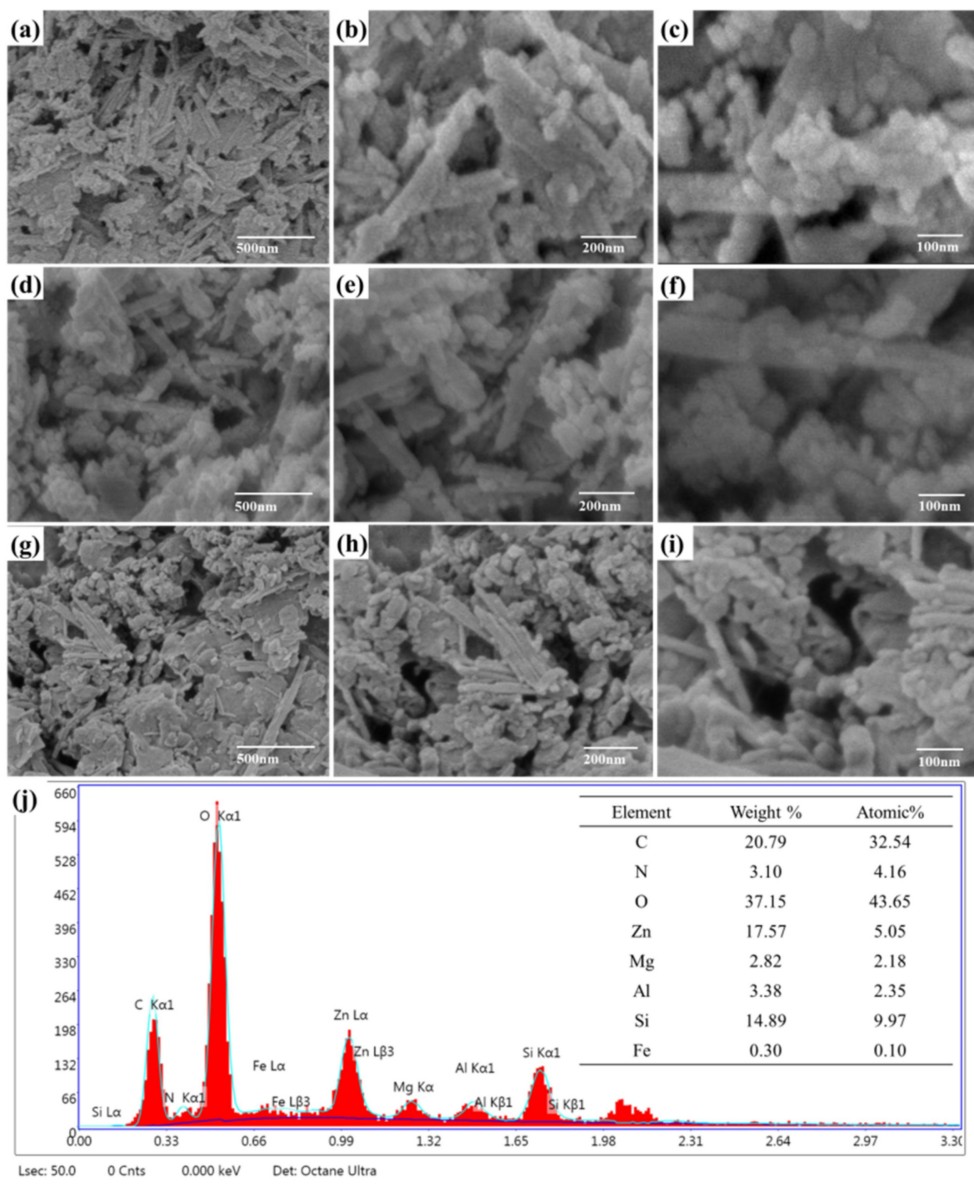

**Figure 1.** FESEM images of (**a**–**c**) ZnO/APT, (**d**–**f**) C–ZnO/APT and (**g**–**i**) MAG/C–ZnO/APT-2.5%, and (**j**) EDX spectrum of MAG/C–ZnO/APT-2.5%.

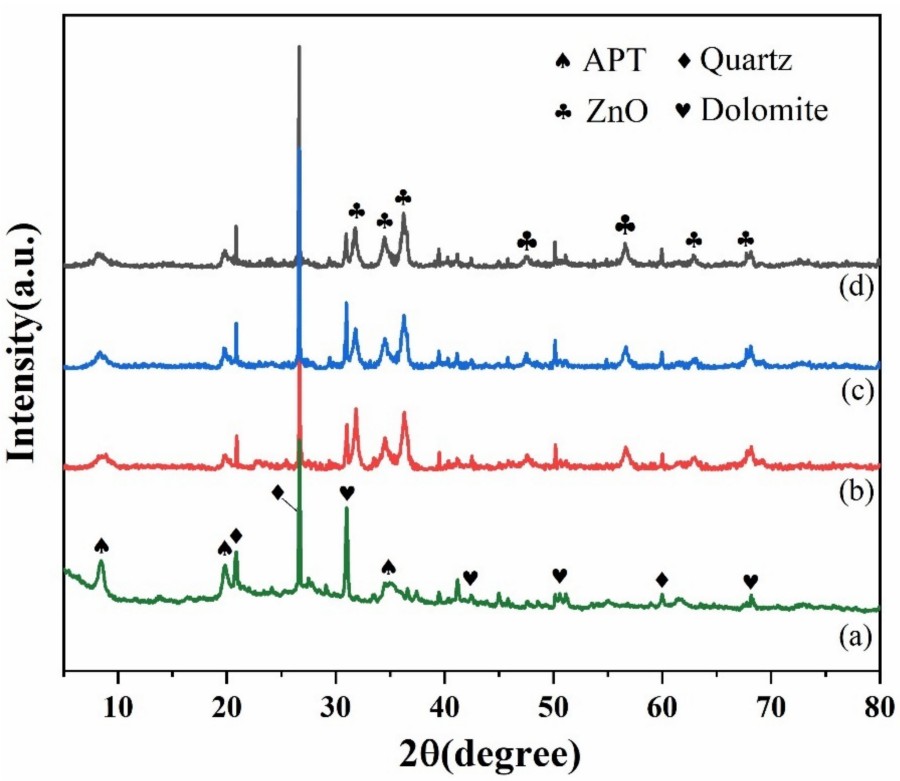

**Figure 2.** XRD patterns of (**a**) APT, (**b**) ZnO/APT, (**c**) C–ZnO/APT-2.5% and (**d**) MAG/C–ZnO/APT-2.5%.

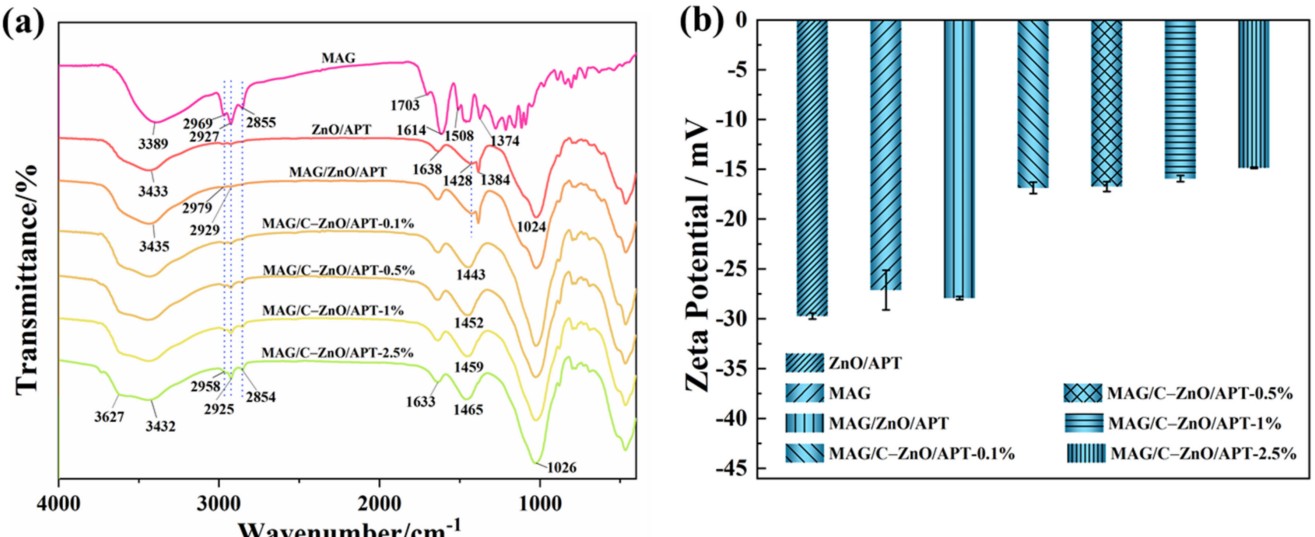

**Figure 3.** FTIR spectra (**a**) and zeta potential (**b**) of MAG, ZnO/APT, MAG/ZnO/APT and MAG/C–ZnO/APT nanocomposites.

The specific surface area, pore volume and pore size of ZnO/APT, C–ZnO/APT and MAG/C–ZnO/APT are given in Table 1. For the MAG/C–ZnO/APT nanocomposites, it can be observed that, when the concentration of CTAB was 0.1%, the composite had the largest total pore volume ($V_{total}$) compared to MAG/C–ZnO/APT treated with more CTAB. This can be attributed to the fact that only a few molecules of CTAB grafted to APT pores and surfaces, as well as the formation of new voids in the aggregates of ZnO/APT [31]. The $V_{total}$ of MAG/C–ZnO/APT with CTAB amounts of 0.5–2.5% were reduced compared with MAG/C–ZnO/APT-0.1%, which possibly resulted from blocking the channels of APT

due to grafting of increasing molecules of CTAB. However, the specific surface area of the nanocomposite increased with increasing dosage of CTAB from 0.1% to 0.5%. This situation was aroused from the dispersion of MAG/C–ZnO/APT nanorods owing to the interparticle hydrophobic interactions contributed by CTAB grafting on ZnO/APT, which avoided the reaggregation of particles [32]. However, for MAG/C–ZnO/APT-0.1%, there was not enough loading of CTAB to achieve this result. Moreover, another study showed that as the dose of CTAB increased continuously (to 1–2.5%), the BET surface area ($S_{BET}$) showed a decreasing trend, which was ascribed to the coverage of CTAB molecules on the pores of APT [33]. In addition, to clarify the changes in the structure of APT by introducing MAG under mechanical action, the corresponding parameters of C–ZnO/APT-2.5% were determined. First, it was found that the $S_{BET}$ of C–ZnO/APT is larger than that of ZnO/APT, which was caused by two reasons. One is that the impurities retained on the channels during the preparation process of ZnO/APT were removed, which is also proven by the increase in its $V_{total}$. Another is related to the fact that APT nanorods loaded with ZnO NPs would aggregate during the calcining process for the preparation of ZnO/APT, and the aggregates could be redispersed to a certain extent during the modification process. After incorporation of MAG into C–ZnO/APT, the $S_{BET}$ increased, which can be explained by the further depolymerization of the APT bundles and some fracture of rod crystals caused by the action of mechanical force leading to an increase in the $S_{BET}$ of the final nanocomposite [34]. The surface coverage by loaded MAG led to a decrease in the $V_{total}$ for MAG/C–ZnO/APT-2.5%. Moreover, it should be noted that the pore size of MAG/C–ZnO/APT nanocomposites with CTAB dosages of 0.5–2.5% was reduced, possibly because the octahedral structure of APT was damaged to some extent [34].

**Table 1.** Pore structural parameters of ZnO/APT, C–ZnO/APT-2.5% and MAG/C–ZnO/APT nanocomposites.

| Samples | $S_{BET}$ (m²/g) | $V_{total}$ (cm³/g) | Pore Size (nm) |
|---|---|---|---|
| ZnO/APT | 48.29 | 0.16 | 13.38 |
| C–ZnO/APT-2.5% | 53.50 | 0.17 | 12.53 |
| MAG/C–ZnO/APT-0.1% | 76.46 | 0.23 | 12.17 |
| MAG/C–ZnO/APT-0.5% | 87.98 | 0.17 | 7.89 |
| MAG/C–ZnO/APT-1% | 83.14 | 0.17 | 8.24 |
| MAG/C–ZnO/APT-2.5% | 74.54 | 0.15 | 8.19 |

TGA/DTG characterization results for ZnO/APT, C–ZnO/APT-2.5% and MAG/C–ZnO/APT nanocomposites are shown in Figure 4. It is found in Figure 4a,b, for ZnO/APT and MAG/C–ZnO/APT, the mass loss below 120 °C can be attributed to the release of free water present on the surface of the composite. The weight losses in the range of 120–400 °C were caused by the release of zeolite water and part of the coordination water of APT except for in MAG/C–ZnO/APT-2.5% [35], for which a loss appearing at 335 °C arosed from the degradation of some CTAB molecules adsorbed on the APT surface. The mass loss around 505 °C for ZnO/APT is due to the release of structural water of APT, but this step was brought forward to around 450 °C for MAG/C–ZnO/APT, which was attributed to the depolymerization of the APT bundles and fracture of the rod crystals, making the structural water easy to remove. This was also contributed by the degradation of MAG and CTAB molecules grafted on ZnO/APT. The third stages of mass loss occurred at 659 °C (ZnO/APT), 700 °C (MAG/C–ZnO/APT-0.1% and MAG/C–ZnO/APT-0.5%) and 690 °C (MAG/C–ZnO/APT-1% and MAG/C–ZnO/APT-2.5%). The collapse of the crystal framework of APT occurred in this temperature range, which led to a large mass loss; the delay for MAG/C–ZnO/APT may have resulted from pre-removal of structural water due to dissociation and breakage of the rod crystals of the nanocomposites. Comparing the mass loss of ZnO/APT, C–ZnO/APT-2.5% and MAG/C–ZnO/APT-2.5%, the successful incorporation of CTAB and MAG on ZnO/APT can be clearly observed from the higher

weight loss. The findings obtained by TGA analysis combined with investigation of specific surface area and the pore structural parameters suggested that the CTAB and MAG molecules were loaded into ZnO/APT effectively; that is, the MAG/C–ZnO/APT nanocomposite was successfully formed in this work.

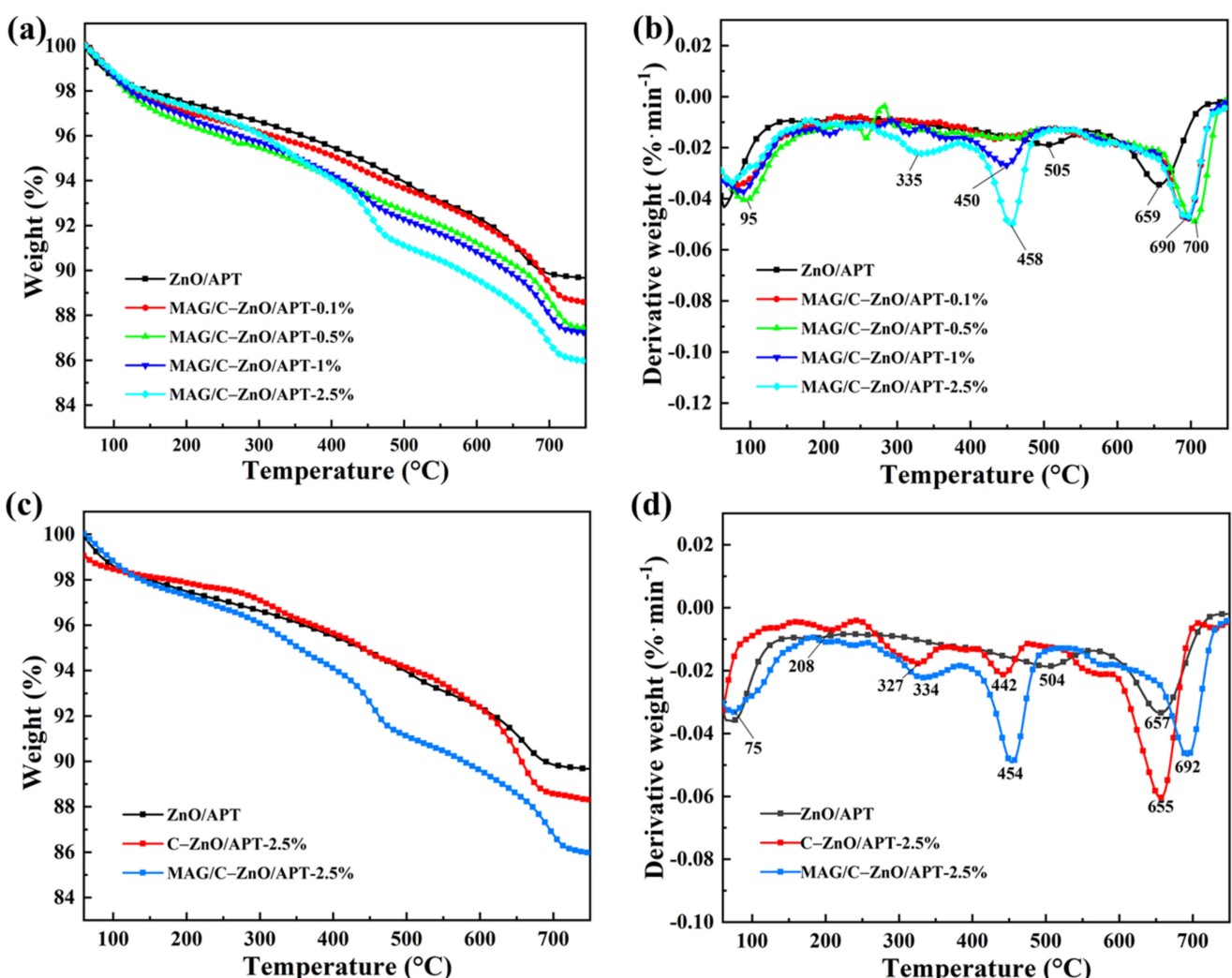

**Figure 4.** TGA (**a**) and DTG (**b**) curves of ZnO/APTand MAG/C–ZnO/APT nanocomposites; TGA (**c**) and DTG (**d**) curves of ZnO/APT, C–ZnO/APT-2.5% and MAG/C–ZnO/APT-2.5%.

### 3.2. Antibacterial Activities of MAG/C–ZnO/APT Nanocomposites

The antibacterial activities of the MAG, ZnO/APT, MAG-ZnO/APT, C–ZnO/APT and MAG/C–ZnO/APT nanocomposites were tested against gram-positive (*S. aureus*) and gram-negative (*E. coli*) bacteria using the agar dilution method. The MIC results of MAG, ZnO/APT, MAG-ZnO/APT, and MAG/C–ZnO/APT are shown in Table S1 and Figure 5, and MIC results of C-ZnO/APT with different amounts of CTAB are shown in Table S2. MAG and ZnO/APT all exhibited antibacterial activities against *S. aureus* and *E. coli*, but they had stronger inhibitory effects toward *S. aureus* than *E. coli*. When ZnO/APT was modified by CTAB and incorporated with MAG, the obtained nanocomposites showed strengthened bacteriostatic effects against *S. aureus* at all CTAB dosages and *E. coli* at CTAB dosages up to 0.5–2.5%. In particular, the MIC values against *S. aureus* and *E. coli* increased to 1 and 0.1 mg/mL, respectively, when 2.5% CTAB was added. Table S3 shows the antibacterial performances of ZnO-based nanocomposites and two organic-inorganic nanocomposites reported in the literature. Compared with ZnO-loaded

nanocomposites with high loading amounts of ZnO compared with that in this study, the MAG/C–ZnO/APT-2.5% obtained in this study exhibited stronger antibacterial activities against *E. coli* and *S. aureus*, suggesting that introduction of MAG and CTAB for modification is an efficient strategy to enhance antibacterial performance. Moreover, the results also can be comparable to the nanocomposites formed by incorporating quaternary ammonium chitooligosaccharides and essential oil to modify ZnO/APT. Furthermore, the antibacterial activities of MAG/C–ZnO/APT-2.5% against *E. faecalis*, *P. aeruginosa*, MRSA and ESBL-*E. coli* were also investigated, and the MIC values are shown in Figure 6. The growth of the four kinds of bacteria was significantly inhibited at the tested concentrations, and the MICs were 0.25, 5, 0.1, and 2.5 mg/mL. The antibacterial activities toward *E. coli* were better than those of MAG, ZnO/APT, MAG-ZnO/APT and C–ZnO/APT, suggesting that incorporation of MAG played a vital role in enhancing the inhibitory actions against gram-negative bacteria.

It can be found from the above antibacterial results that gram-positive bacterial strains seem to be more sensitive to MAG/C–ZnO/APT than gram-negatives. This finding is related to the differences in the cell wall structure of the bacteria, which controls the affinity of different molecules with the cell and the access of the agents to bacterial cells and is particularly associated with the bacteriostatic actions of the nanocomposites. Compared with gram-positive bacteria, gram-negative bacteria have lipopolysaccharide portions at the outer membrane, which are more resistant to harmful substances [36], and MAG with rich hydroxyl groups could directly bind with the proteins in the cell membrane of gram-positive bacteria [37]. These two points may induce the difference in activities of MAG/C–ZnO/APT nanocomposites.

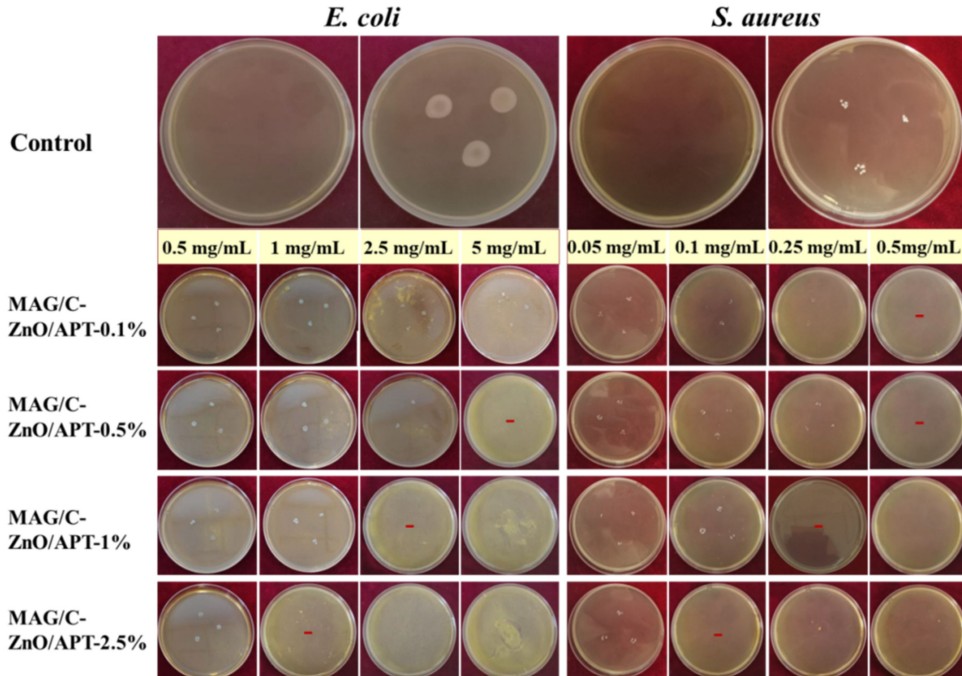

**Figure 5.** MIC values of MAG/C–ZnO/APT nanocomposites toward *E. coli* and *S. aureus*. The red "-" denotes no bacteria in the plates.

The possible antibacterial mechanism of MAG/C–ZnO/APT nanocomposites was proposed as shown in Figure 7. The antibacterial property of ZnO NPs results from reactive oxygen species leading to damage in the cell interior and sustained release of $Zn^{2+}$, which can bind with some groups of proteins to destroy the physiological activity of bacterial cells. After entering the cell, ZnO NPs destroy enzymes in the electron transfer system by binding with the -SH group [35]. MAG exerts inhibitory effects possibly by decreasing the expression of bacterial genes [37,38] and enhancing cell membrane permeability by reacting

with membrane lipids [39]. Therefore, it can be argued that the antibacterial performance can be strengthened via improvement in the contact of ZnO NPs and MAG molecules with bacterial cells [40]. For MAG/C–ZnO/APT nanocomposites, exactly the APT carrier could adsorb or capture the bacteria in particular with modification of CTAB, which moreover improved the affinity between the composite and bacteria [10], making ZnO and MAG attached to the cell membrane of bacteria and play stronger bactericidal actions. More importantly, CTAB has a strong broad-spectrum antimicrobial effect attributed to a disruption of the membrane structure through interaction with the lipid components and the proteins in the cell membrane that are essential to the function of the cells [41–43]. Thus, the enhancement of the inhibition effect of MAG/C–ZnO/APT against gram-negative bacteria may be due to the introduction of CTAB; of course, its other more important role is to firmly fix MAG on the ZnO/APT surface. In brief, the enhanced antimicrobial activity of the obtained nanocomposites was attributed to the synergistic effect of each component in MAG/C–ZnO/APT. such a combination strategy not only obtained synergistically enhanced antibacterial performance but also significantly reduced the dosages of ZnO, which will be more promising as an efficient candidate replacing antibiotics to treat and prevent bacterial infections as well as possibly promote animal growth due to the introduction of MAG that possesses many biological benefits [17].

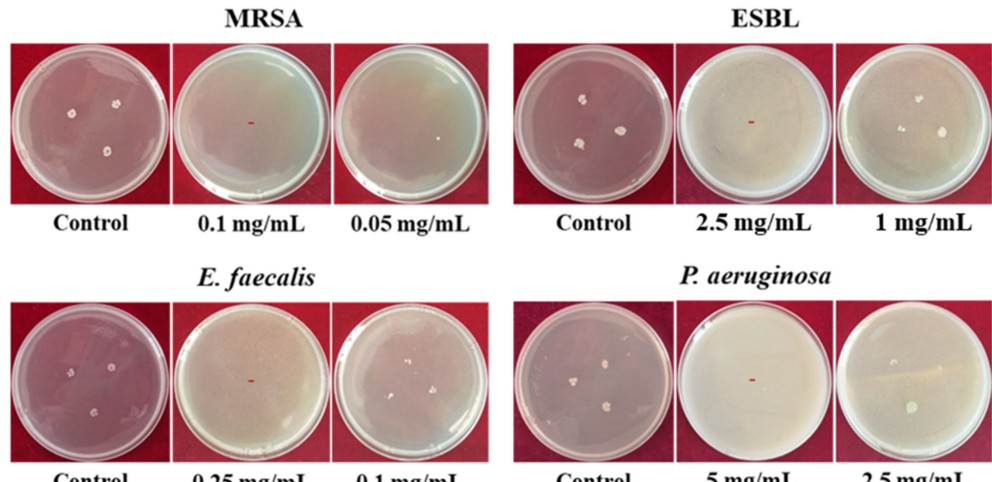

**Figure 6.** MIC values of MAG/C–ZnO/APT-2.5%nanocomposite toward MRSA, ESBL-*E. coli*, *E. faecalis* and *P. aeruginosa*. The red "-" denotes no bacteria in the plates.

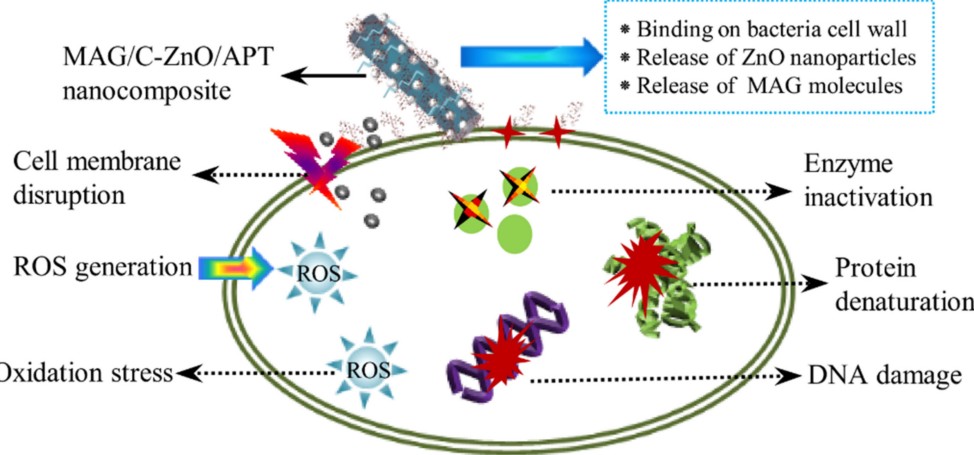

**Figure 7.** Possible mechanisms of MAG/C–ZnO/APT nanocomposites.

### 3.3. Cytotoxicity Test

The cytotoxicity of MAG/C–ZnO/APT-2.5% nanocomposite was evaluated by MTT assays, the results were shown in Figure 8. The MTT results demonstrated concentration-dependent cytotoxicity after exposure to MAG/C–ZnO/APT-2.5% for 24 h. As the sample concentration was 0.625 mg/mL, the nanocomposite exhibited low cytotoxicity. However, with increasing concentration of the sample, the cell viability was dropping, and the $C_{50}$ Values for 24 h incubation was 3.75 mg/mL. Zinc oxide NPs usually have high cytotoxicity on human normal cell lines. Guan et al. assayed the cytotoxicity of ZnO NPs on L-O2 and found that ZnO NPs had concentration-dependent cytotoxicity, and after exposure in 0.1 mg/mL of ZnO NPs, the percentage MTT reduction of L-O2 was 36.70% [44]. In this study, a loading amount of 15% was adopted for ZnO NPs, and under the further introduction of CTAB and MAG, which synergistically enhance antibacterial activities and significantly improve cell survival to a certain extent. Moreover, it has been demonstrated that CTAB usually exhibited high cytotoxicity [45]. The study of Zhang et al. confirmed that the CTAB-encapsulated gold nanorods had higher cytotoxic than the ones undergone further polymer coating and citrate stabilized gold nanospheres [46], which revealed further encapsulation of a CTAB-modified material will help to reduce the cytotoxicity induced by CTAB. Therefore, combinations of the antibacterial agents with low dosage produced a synergistic effect for strengthening the antibacterial capability while reducing the cytotoxicity of both ZnO NPs and CTAB, which is exactly what this study expected. The low cytotoxicity suggested a good biocompatibility for the nanocomposite [47], which will be safe to be used in animal feeding.

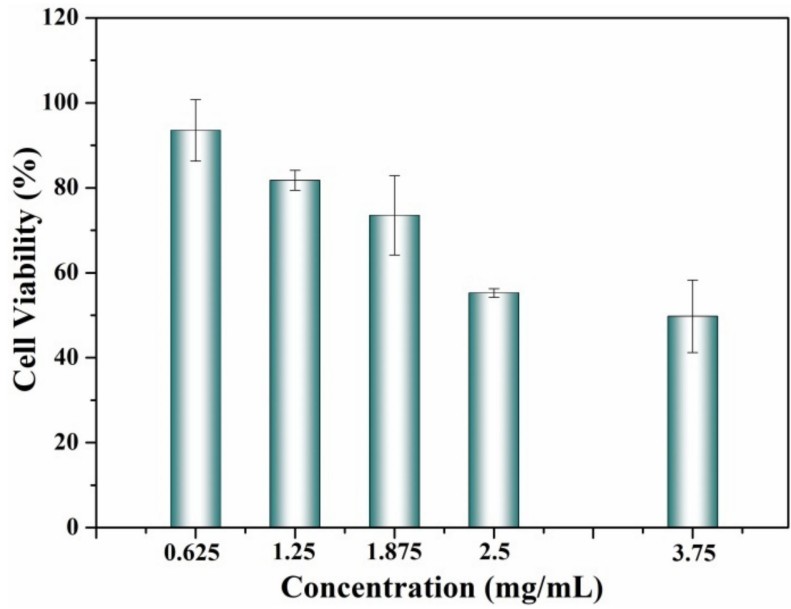

**Figure 8.** Cytotoxicity of MAG/C–ZnO/APT-2.5% nanocomposite on L-O2.

### 4. Conclusions

In this study, MAG/C–ZnO/APT antibacterial nanocomposites were developed by incorporating MAG into the CTAB-modified ZnO/APT under a mechanical milling process. MAG was successfully immobilized on the surface of C–ZnO/APT, where ZnO NPs had a size of 20–50 nm and were uniformly anchored on the surface of APT. It was confirmed that H-bonding and electrostatic interactions play an important role in the incorporation of MAG into C–ZnO/APT. The MAG/C–ZnO/APT nanocomposite with the added amount of CTAB being 2.5% showed good antibacterial activities against *E. coli*, *S. aureus*, *E. faecalis*, *P. aeruginosa*, MRSA and EBSL-*E. coli* and the MIC values were 1, 0.1, 0.25, 5, 0.1, and 2.5 mg/mL, respectively, which was more efficient than MAG, ZnO/APT and C–ZnO/APT.

The enhancement effect can be attributed to the synergistic action of MAG, CTAB and ZnO/APT. Notably, the modification by the cationic surfactant improved the interactions between MAG molecules and ZnO/APT and the introduction of MAG mainly enhanced the sensitivity of the nanocomposites with bacteria. Moreover, the nanocomposite exhibited low cytotoxicity. Therefore, combining natural active ingredients to modify inorganic nanomaterial could be an effective approach to enhance performance and possibly reduce the risk of side effects, which is worthy of further investigation for applications in animal feeding as alternatives to antibiotics.

**Supplementary Materials:** The following supporting information can be downloaded at: https://www.mdpi.com/article/10.3390/min12030364/s1, Figure S1: TEM images and SAED pattern of ZnO/APT. Figure S2: FTIR spectra of MAG, ZnO/APT and C–ZnO/APT nanocomposites with CTAB concentrations of 0.1%, 0.5%, 1% and 2.5%; Figure S3: Zeta potential of C–ZnO/APT nanocomposites with CTAB concentrations of 0.1%, 0.5%, 1% and 2.5%; Table S1: MIC values of MAG, ZnO/APT, and MAG/C–ZnO/APT nanocomposites toward *E. coli* and *S. aureus*; Table S2: MIC values of C–ZnO/APT nanocomposites against *E. coli* and *S. aureus*; Table S3: Comparison of our MAG/C–ZnO/APT nanocomposite with the reported antibacterial nanocomposites; references.

**Author Contributions:** Conceptualization, F.Y. and A.W.; methodology, A.W.; software, Y.S.; validation, A.H., Y.K. and Y.Z.; formal analysis, F.Y. and Y.S.; investigation, F.Y. and Y.S.; resources, all; data curation, F.Y.; writing—original draft preparation, F.Y.; writing—review and editing, A.W. and Y.Z.; visualization, all; supervision, A.W.; project administration, A.W. All authors have read and agreed to the published version of the manuscript.

**Funding:** This research was funded by the Foundation for the Major Projects of the Regional Key Project of the Science and Technology Service of the Chinese Academy of Sciences, China (KFJ-STS-QYZX-086), the Youth Cooperation Fund of Lanzhou Institute of Chemical Physics [HZJJ20-08] and the Natural Science Foundation of Gansu, China (21JR7RA079).

**Data Availability Statement:** The data used to support the findings of this study are available from the corresponding author upon request.

**Conflicts of Interest:** The authors declare no conflict of interest.

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
