# Peer review of "Facile Preparation of Organo-Modified ZnO/Attapulgite Nanocomposites Loaded with Monoammonium Glycyrrhizinate via Mechanical Milling and Their Synergistic Antibacterial Effect"

_minerals, doi:10.3390/min12030364_

Round 1
Reviewer 1 Report
Title: Facile preparation of organo-modified ZnO/attapulgite nanocomposites loaded with monoammonium glycyrrhizinate via mechanical milling and their synergistic antibacterial effect
This manuscript reports the preparation of organo-modified ZnO/attapulgite nanocomposites loaded with monoammonium glycyrrhizinate as an agent antibacterial. The authors carefully conducted the sample preparation to obtain nanocomposites materials, which possessed advantageous structures for the antibacterial activity. Overall, this work is meaningful and interesting, and it will be useful to other researchers in the relevant fields. However, several problems need to resolve, and there are also some mistakes in the manuscript.
The reviewer recommends publication of the present manuscript with minor modifications.
Comment
1-Moreover, there are some mistakes/errors, e.g., “form …." Line 214.It seems that English should be checked.
2-In experimental section: Please do not use abbreviations like "APT" without giving the full name.
3-Another point, The XRD pattern should be revised; please add the patterns of APT and ZnO alone as a reference, as well as FTIR spectra.
4-How did you calculate the particle size? Did you compare with the results obtained using sheerer equation?
5- How the mass loading of ZnO on APT was controlled?
6- Which one played a more important role in the antibacterial activity of composites, MAG or ZnO NPs?
7- Why ZnO-APT was treated using different concentrations of CTAB? What are the roles of Pal of this study?
8- What are the advantages of the preparation method in this study compared with the reported coprecipitation and hydrothermal methods?

Author Response
Comment: This manuscript reports the preparation of organo-modified ZnO/attapulgite nanocomposites loaded with monoammonium glycyrrhizinate as an agent antibacterial. The authors carefully conducted the sample preparation to obtain nanocomposites materials, which possessed advantageous structures for the antibacterial activity. Overall, this work is meaningful and interesting, and it will be useful to other researchers in the relevant fields. However, several problems need to resolve, and there are also some mistakes in the manuscript. The reviewer recommends publication of the present manuscript with minor modifications.
Many thanks for the positive comments and valuable advices on our manuscript.
1-Moreover, there are some mistakes/errors, e.g., “form …." Line 214.It seems that English should be checked.
Reply: Thanks for your suggestion! We have carefully checked and corrected the errors in the revised manuscript, which were highlighted in yellow color.
2-In experimental section: Please do not use abbreviations like "APT" without giving the full name.
Reply: Thanks for your suggestion! We have defined the abbreviations like “APT” and “MAG” in experimental section.
3-Another point, The XRD pattern should be revised; please add the patterns of APT and ZnO alone as a reference, as well as FTIR spectra.
Reply: Thanks for your kind advices. After taking your suggestion seriously, we added the XRD pattern of APT (Please find it in Figure 2), which will provide more information for clarifying the formation of ZnO nanoparticles on APT. In fact, as mentioned in our manuscript, this study focused on the incorporation of MAG into the organo-modified ZnO/APT nanocomposites via a mechanical process for the improvement of the antibacterial activities. ZnO/APT nanocomposite used in this study was prepared following the method in our other study (Hui, A.; Yan, R.; Wang, W.; Wang, Q.; Zhou, Y.; Wang, A. Incorporation of quaternary ammonium chitooligosaccharides on ZnO/palygorskite nanocomposites for enhancing antibacterial activities. Carbohydr. Polym. 2020, 247, 116685), and a more in-depth analysis for ZnO/APT was provided in that published work. Therefore, in this manuscript we provided characterization and properties of ZnO/APT, which was needed to offer reference for C-ZnO/APT and MAG/C-ZnO/APT nanocomposites.
4-How did you calculate the particle size? Did you compare with the results obtained using sheerer equation?
Reply: The particle size of ZnO nanoparticles was identified from the SEM images. We indeed did not compare this result with the size obtained using sheerer equation, which have been complemented in our revised manuscript that was highlighted in yellow color.
5- How the mass loading of ZnO on APT was controlled?
Reply: The precipitation method used for ZnO synthesis is a very mature approach, and which has a high production rate and high purity for the product. Moreover, it can be observed from the SEM images that ZnO nanoparticles practically covered on the surface of APT nanorods and a relative loading of Zn analysis by X-ray fluorescence is 13.56% that is close to theoretical value 15%. In fact, in this work, we expected that a low content of ZnO combined with the modification of CTAB and MAG, ultimately obtaining a performance-enhanced composite antibacterial nanomaterial, so the incorporation of organic agents is the point.
6- Which one played a more important role in the antibacterial activity of composites, MAG or ZnO NPs?
Reply: From the antibacterial results shown in Table SI, S2 and Figure 5, it can be inferred that MAG and ZnO NPs both contribute to the inhibitory activities against bacteria. MAG with rich hydroxyl groups could directly bind with the proteins in the cell membrane of gram-positive bacteria, so incorporation of MAG enhanced the sensitivity of S. aureus to the nanocomposite.
7- Why ZnO-APT was treated using different concentrations of CTAB? What are the roles of Pal of this study?
Reply: In this study, introduction of CTAB to modifying ZnO/APT was to on the one hand enhance the inhibition effect of the final composites against gram-negative bacteria, on the other hand, to firmly fix MAG on the ZnO/APT surface. The different concentrations of CTAB were used to strengthen the inhibitory effect toward bacteria, which can be supported by the results in Table S2. For PAL, firstly we are very sorry about this mistake. Pal is an abbreviation for Palygorskite, which is another name for attapulgite (APT), and we have corrected it in our revised manuscript.
8- What are the advantages of the preparation method in this study compared with the reported coprecipitation and hydrothermal methods?
Reply: In fact, in this manuscript, we prepared the ZnO/APT nanocomposite by a deposition process, which is a simple and widely used method for ZnO synthesis and was used in our other published studies. The novelty involved preparation method is the mechanical milling process used for incorporation of MAG into C-ZnO/APT. The physical method based on milling process is green and ecofriendly due to absence of toxic chemical solvent and reducing of additional hazardous substances compared with the traditional chemical methods.
We really appreciate your valuable and constructive suggestions, and we hope that you are satisfied with our answers and the present manuscript which is obviously improved with your help.
Reviewer 2 Report
In the present study, modified ZnO/APT composites were prepared for antibacterial applications. The composites were characterized by several techniques and their antibacterial properties were investigated against various bacteria and their cytocompatibility was also tested. Overall, the idea is interesting and the data is presented well but the manuscript can be published after addressing the following issues.
- In Figure 1, the ZnO-NPs are not very obvious, it is suggested to supplement TEM analysis to better understand their structures and sizes.
- Figure 1J is EDX spectrum not SAED spectrum, please correct it.
- Section 2.2 “ZnO/PAL….” Do you mean “ZnO/APT”? please correct it.
- Line no. 230 “….no relevant new peaks can be observed…” this statement is misleading. In Figure 2, there are several unexplained diffraction peaks, which need to be explained. Moreover, references should be added for ZnO-NPs diffractions (10.1016/j.ijbiomac.2016.03.044).
- In Figure 3A, there is a distinct peak for ZnO around 400 cm-1, you must describe it (10.1016/j.ijbiomac.2019.03.240).
- Cytotoxicity assay needs references (10.1016/j.colsurfb.2019.110486, 1021/acsami.1c06986).
- Line no. 66 “The antibacterial activity of metal………..” need references (10.3390/polym9120636, 10.1016/j.biotechadv.2021.107856,).
- Add some discussion in Section 3.2 (10.1016/j.compositesb.2020.108208, 10.1016/j.ijbiomac.2018.10.105).
- Some sentences should be revised, for example Line no. 49-53, 118-124.
- Formula of zinc nitrate hexahydrate should be corrected as Zn(NO3)2.6H2
Author Response
Comment: In the present study, modified ZnO/APT composites were prepared for antibacterial applications. The composites were characterized by several techniques and their antibacterial properties were investigated against various bacteria and their cytocompatibility was also tested. Overall, the idea is interesting and the data is presented well but the manuscript can be published after addressing the following issues.
Many thanks for the positive comments and valuable advices on our manuscript.
- In Figure 1, the ZnO-NPs are not very obvious, it is suggested to supplement TEM analysis to better understand their structures and sizes.
Reply: Thanks for your valuable suggestion. We have supplemented the TEM analysis that can be found in section 3.1 in our revised manuscript and Figure S1 in the Supplementary Materials.
- Figure 1J is EDX spectrum not SAED spectrum, please correct it.
Reply: Thanks for the correction. We have revised this error in our revised manuscript.
- Section 2.2 “ZnO/PAL….” Do you mean “ZnO/APT”? please correct it.
Reply: Yes, it is a mistake. We have corrected it in our revised manuscript.
- Line no. 230 “….no relevant new peaks can be observed…” this statement is misleading. In Figure2, there are several unexplained diffraction peaks, which need to be explained. Moreover, references should be added for ZnO-NPs diffractions (10.1016/j.ijbiomac.2016.03.044).
Reply: We have revised the statement of XRD, and noted the peaks of the associated minerals including quarts and dolomite in Figure 2. Moreover, the reference was added.
- In Figure 3A, there is a distinct peak for ZnO around 400 cm-1, you must describe it (10.1016/j.ijbiomac.2019.03.240).
Reply: Yes, you are right. A distinct peak for ZnO usually can be found around 400 cm-1. However, in the spectrum of ZnO/APT, the peaks in the range of 400-500 cm-1 related to the asymmetric stretching vibration of O-Si-O in APT would interfere with the peak of ZnO. In fact, in our manuscript the SEM-EDS、XRD and TEM analysis can fully prove the formation of ZnO nanoparticles on APT surface.
- Cytotoxicity assay needs references (10.1016/j.colsurfb.2019.110486,1021/acsami.1c06986).
Line no. 66 “The antibacterial activity of metal………..” need references (10.3390/polym9120636,10.1016/j.biotechadv.2021.107856,).
Reply: Thanks for your recommendation. We have read up carefully and cited one of them.
- Add some discussion in Section 3.2 (10.1016/j.compositesb.2020.108208,
10.1016/j.ijbiomac.2018.10.105).
Reply: In fact, we have well discussed the antibacterial performance of MAG/C-ZnO/APT nanocomposites and the possible synergistic mechanism. Thanks for your recommendation of references, and we have read up carefully and cited one of them.
- Some sentences should be revised, for example Line no. 49-53, 118-124.
Formula of zinc nitrate hexahydrate should be corrected as Zn(NO3)2.6H2O
Reply: Thanks for your correction. According your suggestions, we have revised the relative content in our revised manuscript.
We really appreciate your valuable and constructive suggestions, and we hope that you are satisfied with the present manuscript which is obviously improved with your help.
Reviewer 3 Report
The manuscript title is "Facile preparation of organo-modified ZnO/attapulgite nanocomposites loaded with monoammonium glycyrrhizinate via mechanical milling and their synergistic antibacterial effect". In the study, a MAG/C-ZnO/APT composite was obtained by a chemical multistage method. Introduced natural substances have an antibacterial effect and together have a synergistic effect. The proposed material is of considerable interest as an antiviral/antibacterial substance, and the article itself is of considerable interest to manufacturers and researchers. In addition, the article is a successful cooperation of the scientific teams participating in the work. The manuscript is very interesting, and the results are significant, but the description of the experiments, some methodological features need to be improved.
- Could you please add to the introduction the intended use of the resulting composite material? For example, ointments, antiseptic dressings, etc.
- Why normal cell lines L-O2 was used for MTT test?
- Unfortunately, SEM images are not very informative for the detection of organic substances as MAG/C. It would be nice to demonstrate EDA mapping for MAG/C-ZnO/APT samples. Also, it would be nice, if SEM-images would the same magnifications (a, d, g) and (b, e, h), (c, f, i).
- SAED spectra would be present for each stage of synthesis process and would be better discussed.
- Some peaks on XRD patterns are nor identified, please, explain them.
- Is it possible to set the percentage of each of the active components in the composite?
- Please, explain, how pore structural parameters were calculated.
- When conducting the MTT test, it would be important to determine the effect of the carrier ZnO/APT, C-ZnO/APT and active substances MAG separately. That can have a joint effect with carrier and the active substances that are stated with the manuscript. It would be good if the authors were able to determine the synergistic effect of the substances in the composite. What substances were used for negative and positive probes.
Author Response
Comment: The manuscript title is "Facile preparation of organo-modified ZnO/attapulgite nanocomposites loaded with monoammonium glycyrrhizinate via mechanical milling and their synergistic antibacterial effect". In the study, a MAG/C-ZnO/APT composite was obtained by a chemical multistage method. Introduced natural substances have an antibacterial effect and together have a synergistic effect. The proposed material is of considerable interest as an antiviral/antibacterial substance, and the article itself is of considerable interest to manufacturers and researchers. In addition, the article is a successful cooperation of the scientific teams participating in the work. The manuscript is very interesting, and the results are significant, but the description of the experiments, some methodological features need to be improved.
Many thanks for the positive comments and valuable advices on our manuscript.
- Could you please add to the introduction the intended use of the resulting composite material? For example, ointments, antiseptic dressings, etc.
Reply: Thanks for your suggestion! We expected this nanocomposite can be promising to be used in animal feedings for preventing bacterial infection and promoting growth of animals. The relative content was presented in Introduction section and has been further improved according to your suggestion.
- Why normal cell lines L-O2 was used for MTT test?
Reply: L-O2 is a normal human hepatocyte, which is more sensitive to toxic effects. Thus in this study in order to evaluate the toxic effect of MAG/C-ZnO/APT on normal cells, L-O2 was selected for MTT test. In fact, L-O2 also is a commonly used cell line for cytotoxicity evaluation.
- Unfortunately, SEM images are not very informative for the detection of organic substances as MAG/C. It would be nice to demonstrate EDA mapping for MAG/CZnO/APT samples. Also, it would be nice, if SEM-images would the same magnifications (a, d, g) and (b, e, h), (c, f, i).
Reply: Thanks for your kind advices. For the detection of organic substances we obtained the EDX spectra using spot scanning considering that it is difficult to determine whether active ingredients are loaded onto attapulgite from the surface scanning, so we just provided the EDX pattern for the nanocomposite. According your suggestion, we have updated the SEM images, which can be found in Figure 1.
- SAED spectra would be present for each stage of synthesis process and would be better discussed.
Reply: Yes, you are right. The EDX analysis for each stage such as ZnO/APT and C-ZnO/APT, will well present the distribution of ZnO nanoparticles and organic groups. Although the EDX analysis for each stage was not provided in the present manuscript, some other means were presented to achieve the equivalent purpose including the supplementary information. ZnO/APT prepared in this study is following the method in our other published work (Hui et al. Carbohydr. Polym. 2020, 247, 116685), in which a detailed study of ZnO/APT was presented. According your suggestion, we added the TEM images and SAED analysis in the Supplementary materials, which offers more information of ZnO NPs loaded on APT nanorods. For the organo-modification, the FTIR and zeta potential analysis as well as the antibacterial activity were provided to clarify the interfacial interactions of MAG and C-ZnO/APT and the synergistic effect for enhanced antibacterial performance that is the role of organo-modification.
- Some peaks on XRD patterns are nor identified, please, explain them.
Reply: These peaks are related to the associated minerals in APT. In order to state the problem more clearly, we have added the XRD pattern of APT and noted the peaks of the associated minerals. Please find in Figure 2 and Section 3.1 that was highlighted in yellow color.
- Is it possible to set the percentage of each of the active components in the composite?
Reply: It is possible but very complicated task, we can try in our next study, maybe will achieve more favorable results. In fact, in this work, we have considered the facts of each of the active components. The low loading of ZnO was selected is based on our previous study about ZnO/APT nanocomposites, where a low loading is in favor of a smaller size of ZnO NPs due to their dispersion on APT nanorods. 0.25% of MAG was used based on a pre-experiment of MAG/ZnO/APT, and when the addition amount of MAG up to 0.25% an enhanced antibacterial activity was obtained that was shown in Table SI.
- Please, explain, how pore structural parameters were calculated.
Reply: The SBET values were from the BET method. Vtotal values were obtained from the single point adsorption total pore volume of pores less than 83.3769 nm diameter at P/Po = 0.976219009, and the pore sizes were calculated from the adsorption average pore width (4 Vtotal /SBET).
- When conducting the MTT test, it would be important to determine the effect of the carrier ZnO/APT, C-ZnO/APT and active substances MAG separately. That can have a joint effect with carrier and the active substances that are stated with the manuscript. It would be good if the authors were able to determine the synergistic effect of the substances in the composite. What substances were used for negative and positive probes.
Reply: We are very pleased with your suggestion. In fact, the further study for this nanocomposite is intended to clarify the mechanism of the synergistic effect of the nanocomposite involved the antibacterial performance and cytotoxicity, and in-depth and detailed study of the toxic effects of each component will help to clarify the mechanism. In this study, the cytotoxicity was evaluated just to exhibit the low cytotoxicity of MAG/C-ZnO/APT nanocomposite, because the main innovation presented in this manuscript would be the improvement of antibacterial performance by incorporation of MAG into the organo-modified inorganic nanomaterial.
The negative probe was the LB agar plate without antibacterial materials and bacteria, and the positive one referred the LB agar plate without antibacterial materials but inoculated bacteria.
We really appreciate your valuable and constructive suggestions, and we hope that you are satisfied with the present manuscript which is obviously improved with your help.
Reviewer 4 Report
Manuscript:
Title: Facile preparation of organo-modified ZnO/attapulgite nano-composites loaded with monoammonium glycyrrhizinate via mechanical milling and their synergistic antibacterial effect.
The content of the article titled: "Facile preparation of organo-modified ZnO/attapulgite nano-composites loaded with monoammonium glycyrrhizinate via mechanical milling and their synergistic antibacterial effect" has successfully researched the antibacterial nanomaterial MAG/C-ZnO/APT. The results obtained are very interesting which, at present, cannot be published unless the authors make very large edits to the content of the article, through the following recommendations:
- The results got at present, have not clarified the nature of the problem to be studied, the authors need to analyze and clarify the following issues:
-It is necessary to clarify the nature of the MAG incorporation into the ZnO/APT carrier, why the hybrid author chose this combination
-What causes the chemical interaction to lead to the material's good antibacterial ability and resistance to E. coli, S. aureus, E. faecalis, P. aeruginosa, MRSA and EBSL-E. It is necessary to clarify the nature of this chemical interaction.
-The results got should be directly compared with other results got to confirm the accuracy of the results.
- In the introduction, the author said: "The ability to enhance susceptibility to gram-positive bacteria, and demonstrated the interaction of MAG molecules with the carrier", should be indicated. In the content of the introduction was poorly written, the summary and the conclusion did not highlight the meaning of the article content. The authors need to revise the entire content and add the results obtained by other methods to compare and clarify the content of the article to enhance the attractiveness and confirm the accuracy of the results got.
3.Why did the authors choose to use ZnO/APT with the ratio C-ZnO/APT-2.5% and MAG/C-ZnO/APT-2.5%. So use C- and MAG with ratios > 2.5% then what is going to happen ?. In addition, the investigation of materials in different temperatures, the author needs to clarify this issue. More specifically, at what temperature is the material the author is studying and with such a high temperature, does the applicability in the biomedical industry have any effect?
- Many passages in the materials and methods section are duplicated with previously published articles, so the authors are requested to correct them to avoid plagiarism and check by Turnitin anti-plagiarism software.
- The authors should check the entire manuscript, correct all English grammar errors or use a professional English editing service.
Wish the author team success, with this useful work.

Author Response
Comment: The content of the article titled: "Facile preparation of organo-modified ZnO/attapulgite nano-composites loaded with monoammonium glycyrrhizinate via mechanical milling and their synergistic antibacterial effect" has successfully researched the antibacterial nanomaterial MAG/C-ZnO/APT. The results obtained are very interesting which, at present, cannot be published unless the authors make very large edits to the content of the article, through the following recommendations:
Many thanks for your valuable advices on our manuscript.
- The results got at present, have not clarified the nature of the problem to be studied, the authors need to analyze and clarify the following issues:
-It is necessary to clarify the nature of the MAG incorporation into the ZnO/APT carrier, why the hybrid author chose this combination
Reply: Yes, indeed there was not much in-depth study for the nature of MAG. In fact, we have provided the reported information of MAG in introduction that MAG is a derivative of glycyrrhizic acid (GA) obtained from the roots of licorice, having remarkable bacterial inhibition and other beneficial pharmacological activities, such as anti-inflammatory activity, antiviral activity, autoxidation function and anti-gastric ulcer effects, and recently have been widely used in the pharmaceutical and food industries. Thus in this study we expected that by incorporating this natural ingredients to modify ZnO/APT could achieve enhanced antibacterial performance. Plant-derived natural active components have great potential in biomedical applications due to their many advantages, such as low toxicity, extensive resources and lack of pollution to the environment, which have become hot candidates in the antimicrobial field, especially in post-antibiotic era. ZnO as an effective agent for antibiotic substance could achieve very good results in preventing diarrhea and promoting growth in animals and is the most widely studied and applied inorganic antibacterial agent at present. However, high dose addition of ZnO could cause Zn toxicity, Zn accumulation in the environment and even the development of drug-resistance, thus improvement of its antibacterial performance and Zn reduction have become a direction of the development of ZnO-based antimicrobials. APT, as a natural nanoscale clay mineral, exhibits favourable carrier performance due to their unique rod-shaped morphology and rich surface active groups. Our previous studies have demonstrated that APT could act as a good support for dispersion of metal and metal oxide nanoparticles to enhance their relative properties (Hui et al. Carbohydr. Polym. 2020, 247, 116685; Wang et al. J. Nanopart. Res. 2014, 16, 2281). Therefore, in this work, we aim to further enhance the performance of ZnO/APT by introducing MAG, the obtained antibacterial material may be promising for the management of animal health.
In addition, in this study, the FTIR analysis and antibacterial activity evaluation of MAG were conducted to clarify the possible interactions of MAG and C-ZnO/APT as well as the possible synergistic effect for enhancement of the antibacterial properties of the final nanocomposites.
According to your kind suggestion, we will further clarify the nature of the MAG in our next studies, which will contribute to clarification of the antibacterial mechanism of MAG-based composite antibacterial materials. In addition, we have perfected the introduction section in our revised manuscript.
-What causes the chemical interaction to lead to the material's good antibacterial ability and resistance to E. coli, S. aureus, E. faecalis, P. aeruginosa, MRSA and EBSL-E. It is necessary to clarify the nature of this chemical interaction.
Reply: In our manuscript, the FTIR and zeta potential analysis demonstrated that MAG could be immobilized on C-ZnO/APT by forming H-bonding and electrostatic interactions, so we are very sorry for our mistake of description in the Abstract and we have corrected the statements in the Abstract in our revised manuscript. The good antibacterial activities exhibited by the obtained MAG/C-ZnO/APT nanocomposite can be attributed to the synergistic effects of MAG, CTAB and ZnO/APT, such as the charge change induced by CTAB-modification that improved the contact and interface interaction between the material and bacteria. The details for the synergistic enhancement of antibacterial activities were presented in section 3.2 in our manuscript.
-The results got should be directly compared with other results got to confirm the accuracy of the results.
Reply: Yes, you are right. We have added relative contents in our revised manuscript, which were shown in Table S3. In fact, the results obtained in this manuscript were all repeated, such as the MIC test was repeated at least three times.
- In the introduction, the author said: "The ability to enhance susceptibility to gram-positive bacteria, and demonstrated the interaction of MAG molecules with the carrier", should be indicated. In the content of the introduction was poorly written, the summary and the conclusion did not highlight the meaning of the article content. The authors need to revise the entire content and add the results obtained by other methods to compare and clarify the content of the article to enhance the attractiveness and confirm the accuracy of the results got.
Reply: Thanks for your advices. According to your suggestion, we have made much optimization and revision in the Introduction, and added the reported results obtained by other methods to compare and clarify the performance obtained in this study, which is highlighted in yellow color in Introduction section and Section 3.2 and 3.3.
- Why did the authors choose to use ZnO/APT with the ratio C-ZnO/APT-2.5% and MAG/C-ZnO/APT-2.5%. So use C- and MAG with ratios > 2.5% then what is going to happen ?. In addition, the investigation of materials in different temperatures, the author needs to clarify this issue. More specifically, at what temperature is the material the author is studying and with such a high temperature, does the applicability in the biomedical industry have any effect?
Reply: Thanks for your suggestion! In fact, 0.25% of MAG was used based on a pre-experiment of MAG/ZnO/APT, and when the addition amount of MAG up to 0.25% an enhanced antibacterial activity was obtained that was shown in Table SI. For CTAB, we have considered a more amount of CTAB, but we found when CTAB amount increased to 5%, the MIC values of MAG/C-ZnO/APT-5% maintained same to that of MAG/C-ZnO/APT-2.5%, which may be caused by the effective loading of CTAB. Moreover, CTAB as a surfactant has been demonstrated to have greater biological toxicity at a high dosage. So an amount range of 0.15-2.5% was used for modification of ZnO/APT in our study. In addition, all the samples were investigated at room temperature in this study. In view of your suggestion, we will further study the effect of the temperature on performance of the nanocomposites, which will be interesting.
Figure R1. MIC test for MAG/C-ZnO/APT-5%
- Many passages in the materials and methods section are duplicated with previously published articles, so the authors are requested to correct them to avoid plagiarism and check by Turnitin anti-plagiarism software.
Reply: Very thanks for your kind suggestion! We have revised the materials and methods section.
- The authors should check the entire manuscript, correct all English grammar errors or use a professional English editing service.
Reply: Thanks for your valuable advices. We have carefully revised the manuscript and also checked and revised the English according to your suggestions, which were highlighted in our revised manuscript.
Wish the author team success, with this useful work.
We really appreciate your valuable and constructive suggestions, and we hope that you are satisfied with the present manuscript which is obviously improved with your help.
Round 2
Reviewer 4 Report
The revised draft content of the authors at the present time has met the requirements of the reviewer, publication accepted. So, the reviewer has only added small suggestions, recommended authors re-checked the entire grammar style and English errors.
Congratulations to the author team's success with this useful project.
